# SARS-COV-2 antibody responses to AZD1222 vaccination in West Africa

Real-world data on vaccine-elicited neutralising antibody responses for two-dose AZD1222 in African populations are limited. We assessed baseline SARS-CoV-2 seroprevalence and levels of protective neutralizing antibodies prior to vaccination rollout using binding antibodies analysis coupled with pseudotyped virus neutralisation assays in two cohorts from West Africa: Nigerian healthcare workers ($n = 140$) and a Ghanaian community cohort ($n = 527$) pre and post vaccination. We found 44 and 28% of pre-vaccination participants showed IgG anti-N positivity, increasing to 59 and 39% respectively with anti-receptor binding domain (RBD) IgG-specific antibodies. Previous IgG anti-N positivity significantly increased post two-dose neutralizing antibody titres in both populations. Serological evidence of breakthrough infection was observed in 8/49 (16%). Neutralising antibodies were observed to wane in both populations, especially in anti-N negative participants with an observed waning rate of 20% highlighting the need for a combination of additional markers to characterise previous infection. We conclude that AZD1222 is immunogenic in two independent West African cohorts with high background seroprevalence and incidence of breakthrough infection in 2021. Waning titres post second dose indicates the need for booster dosing after AZD1222 in the African setting despite hybrid immunity from previous infection.

Despite highly effective vaccines, SARS-CoV-2 transmission continues. Variants of concern (VOC), likely arising within chronically infected patients[1] and demonstrating both immune escape and increased infectivity[2–9], have compromised protective effects of two-dose vaccines such as BNT162b2 and AZD1222 in the context of suboptimal vaccine coverage/waning of immune responses[2]. Although a third dose with mRNA vaccination is able to rescue neutralisation against Omicron in the short term[2,7], waning has been documented in vulnerable individuals. Fourth doses increase immune responses and are being implemented in some countries for higher risk populations[10].

Across the African region, vaccine rollout has been heterogenous with 16% of total eligible population completing vaccination and 1.3% receiving a booster dose. Serum neutralisation in vitro correlates with protection against SARS-COV-2 infection in clinical studies[11]. However, vaccine rollout in African countries is impaired by paucity of neutralisation data and vaccine efficacy data for VOCs; in particular, there

are no published data on vaccine-elicited neutralising antibody responses for AZD1222−the world's most widely used vaccine currently in African populations following scale up. AZD1222 is a chimpanzee adenovirus-vectored vaccine (ChAdOx1) based on the SARS-CoV-2 spike protein. Adenoviral vectored vaccines generate lower neutralising antibody responses in general compared to mRNA vaccines[2,3,12,13]. T cell responses across both platforms appear to be robust and well preserved, as is protection from severe disease and death[14–16]. Given the lower neutralisation titres, real-world data in settings without access to boosting with mRNA vaccines is particularly important given the continual emergence of new immune evasive variants with varying degrees of severity.

COVID-19 Vaccine rollout across west African populations in Ghana and Nigeria has been characterised by significant disparities despite vaccines from the COVAX facility led by the Coalition for Epidemic Preparedness Innovations (CEPI), Gavi and World Health

✉ e-mail: pm685@cam.ac.uk; phillips@kccr.de; tundesalako@hotmail.com; rkg20@cam.ac.uk

Organization (WHO). Vaccines were first available in February and March 2021 in Ghana and Nigeria respectively. Here, we measured (i) baseline SARS-CoV-2 seroprevalence and levels of protective antibodies prior to vaccination rollout using both flow cytometric based analysis of binding antibodies to nucleocapsid (N), coupled with virus neutralisation approaches and (ii) neutralising antibody responses to VOCs in two West African cohorts prior to vaccination, and after two doses of AZD1222 vaccine administered between June and July 2021 in Lagos, Nigeria and between May to June 2021 in Kumasi, Ghana.

## Results

### Evidence of prior SARS-COV-2 infection by binding antibodies and neutralisation

Our study population in Lagos, Nigeria who received at least one dose of the AZ1222 vaccine comprised of 140 participants with a median age of 40 (inter-quartile range: 33, 48), 73 (52%) of whom were male. In order to analyse the proportion of participants in this urban population previously exposed to SARS-COV-2, we tested all baseline samples ($n = 140$) for anti-N IgG using a flow cytometry based assay[17] and found 62/140 participants were positive prior to administration of first vaccine dose, demonstrating 44% SARS-CoV-2 anti-N IgG seroprevalence at baseline prior to vaccination. 21/78 (27%) anti-N IgG negative subjects were additionally positive for anti-RBD IgG prior to vaccination—yielding a total seroprevalence of 83/140 (59%) in Lagos, Nigeria. Our study population from Kumasi, Ghana, enroled prior to vaccination comprised of 527 participants with a median age of 33 (inter-quartile range: 25, 47), 295 (56%) of whom were males. We tested all baseline samples ($n = 527$) for anti-N IgG by flow cytometry[17] and found 147 participants were positive, demonstrating 28% SARS-CoV-2 anti-N IgG seroprevalence at baseline prior to vaccination. Using RBD positivity, we found additional 57/382 (15%) anti-N negative participants were positive for anti-RBD IgG prior to vaccination, indicating a previously waned anti-N IgG and exposure proportion of 204/527 (39%).

To explore the phenomenon of prior exposure in the context of waned N antibody further we used PV neutralisation assays as described previously[18]. Baseline neutralising GMT (geometric mean titre) of ID50s in the Nigerian study population when stratified by anti-N status was 431 vs 47 in IgG anti-N positive and negative participants respectively, suggestive of the presence of neutralizing antibodies against SARS-CoV-2 in subjects negative for SARS-CoV2 anti-N Ab prior to vaccination. Of the 24 individuals anti-N Ab negative at baseline, 12/24 had an $ID_{50}$ above the cut-off of 20. In these individuals, binding antibodies to S were also detectable, and neutralisation correlated with IgG anti-S and IgG anti-RBD levels ($r = 0.71$ and $r = 0.73$) respectively indicating prior infection in at least half of those who were N Ab negative at baseline (Supplementary Fig. 1a). In the Ghanaian population, the baseline GMT of ID50s against WT PV in the study population when stratified by anti-N status was 106 vs 45 in IgG anti-N positive and negative participants respectively, again suggestive of the presence of neutralizing antibodies against SARS-CoV-2 in subjects negative for SARS-CoV2 anti-N Ab prior to vaccination. Of the 32 individuals anti-N Ab negative at baseline, 15/32 had an $ID_{50}$ above the cut-off of 20. In these individuals, binding antibodies to S were also detectable, and neutralisation moderately correlated with IgG anti-S and IgG anti-RBD levels ($r = 0.57$ and $r = 0.55$) respectively, also indicating prior infection in at least half of those who were N Ab negative at baseline (Supplementary Fig. 1b). Taken together these data suggest that anti-N underestimates the true prevalence of prior infection by ~50%. Of note 3/62 (4.8%) and 10/147 (6.8%) of anti-N positive participants were both anti-RBD and anti-S negative from the Nigerian and Ghanaian populations, respectively.

### Waning of N antibody and reinfections over time

12/49 (24.5%) became anti-N IgG negative at 1 month post second-dose and no further participants lost anti-N IgG positivity between 1- and 3-months post second-dose. Of note, one subject who was anti-N IgG positive at baseline became anti-N IgG negative at 1 month and then became anti-N IgG positive 3-month post second-dose with a 7-fold increase in anti-N IgG titres between 1 month and 3 months post second-dose—strongly suggestive of re-infection. In Ghanaian participants we similarly observed loss of IgG anti-N in 7 of 45 (15.6%) participants with follow-up serum samples at 1 month post vaccination.

### Longitudinal neutralising and binding antibody responses following vaccination

Of the 140 Nigerian participants recruited (Fig. 1), 49 had plasma samples available at baseline prior to vaccination and at two follow-up time points post vaccination for neutralisation assays (Table 1). Median age was 39 (31, 46) and 47% were male. Half of the participants, 25/49 (51%) were IgG anti-N positive at baseline, and the GMT of neutralizing antibodies associated with 50% neutralisation against WT (Wu-1 D614G) PV across the entire study population was 145 ± 4.5 (GMT ± s.d) (Table 1; Fig. 2a) with significantly lower titres observed against the Delta and Omicron variants with GMT titres 75 ± 3.6 (GMT ± s.d) and 55 ± 3.0 (GMT ± s.d) ($p = 0.0001$ and $p < 0.0001$) respectively. Amongst Ghanaian participants, of the 527 participants recruited (Fig. 1), 45 had plasma samples available at baseline prior to vaccination and at 2-month post vaccination for neutralisation assays (Table 1). Almost one-third of participants were IgG anti-N positive at baseline, and the GMT of neutralizing antibodies associated with 50% neutralisation ($ID_{50}$) against WT PV across the entire study population was 57 ± 3.0 (GMT ± s.d) (Table 1; Fig. 2a) with significantly lower titres observed against the Delta and Omicron variants with GMT titres 37 ± 2.4 (GMT ± s.d) and 29 ± 1.8 (GMT ± s.d) ($p < 0.001$ and $p < 0.001$) respectively.

Overall, neutralizing Ab titre to WT 1 month after second dose in Nigerian participants was 2579 ± 4.2 (GMT ± .s.d). As expected, lower neutralisation titres were observed against the Delta [549 ± 3.7 (GMT ± .s.d); ($p < 0.0001$)] and Omicron variants [269 ± 3.4 (GMT ± .s.d); $p < 0.0001$] at 1 month, representing a fold reduction of 4.7 and 9.6 respectively (Fig. 2). The GMT for Delta and Omicron was only around 100, nearly a log lower in comparison to WT (Fig. 2). Positive anti-N IgG Ab status at baseline was associated with significantly higher titres of neutralizing antibodies following vaccination across all tested VOC (Fig. 2). Importantly, those with anti-N Abs present at baseline did not experience waning of responses between months one and three post-second dose (Supplementary Fig. 2).

In Ghana, geometric mean neutralizing Ab titre to WT PV 2 month after second dose was 1049 ± 6.7(GMT ± .s.d). Lower levels of neutralisation were observed against the Delta [453 ± 7.4 (GMT ± .s.d); ($p < 0.0001$)] and Omicron variants [95 ± 5.3 (GMT ± .s.d); $p < 0.0001$] at 2 month, representing a fold reduction of 2.3 and 11.0-fold respectively (Fig. 2).

As observed in the Lagos population, positive anti-N IgG Ab status at baseline was associated with significantly higher titres of neutralizing antibodies following vaccination across tested VOC except the Omicron variant (Fig. 2). When we compared neutralisation in N negative participants post second dose, the GMT for WT were: 1423 ± 3.9 (GMT ± .s.d) in Nigerian participants and 773 ± 7.4 (GMT ± .s.d) in Ghanaian participants respectively.

### Waning of neutralising responses

Overall, there was no decline in neutralising antibody titres at 3 months for WT, Delta, or and Omicron compared to 1 month post vaccination in Nigerian participants (Fig. 2a). By contrast, when data were stratified by anti-N IgG status at any time point, there was a significant decline in neutralisation between 1 month and 3 months post second-dose across all variants tested for participants who were N antibody negative throughout (Figs. 2 and 3, $p = 0.04$). The GMT in these individuals for

Delta and Omicron was ~100, nearly a log lower in comparison to WT (Fig. 2B). Participants with anti-N Abs present at baseline did not experience waning of responses between months 1 and 3 post second dose (Supplementary Fig. 2a and c), despite frequent loss of N antibody over time (Supplementary Fig. 3). When we examined binding antibodies over time in the group as a whole, we saw very small decreases for Wu-1 and Omicron Spike IgG but not for Wu-1 RBD (Supplementary Fig. 4a). When data were analysed for those who were anti-N negative, waning of binding antibodies was more evident (Supplementary Fig. 4b, Supplementary Fig. 5). We were not able to

assess waning due to absence of samples beyond 1 month post second dose in Ghana.

## Vaccine breakthrough infection

To evaluate the proportion of participants with vaccine breakthrough infection after two doses of AZD1222 vaccine, we tested anti-N IgG in subjects who were anti-N IgG negative at baseline ($n = 78$) and became positive between 1- and 3-months post second dose and found 7/49 (14%) with de novo infection, with one additional participant demonstrating both reinfection and breakthrough infection to yield a total

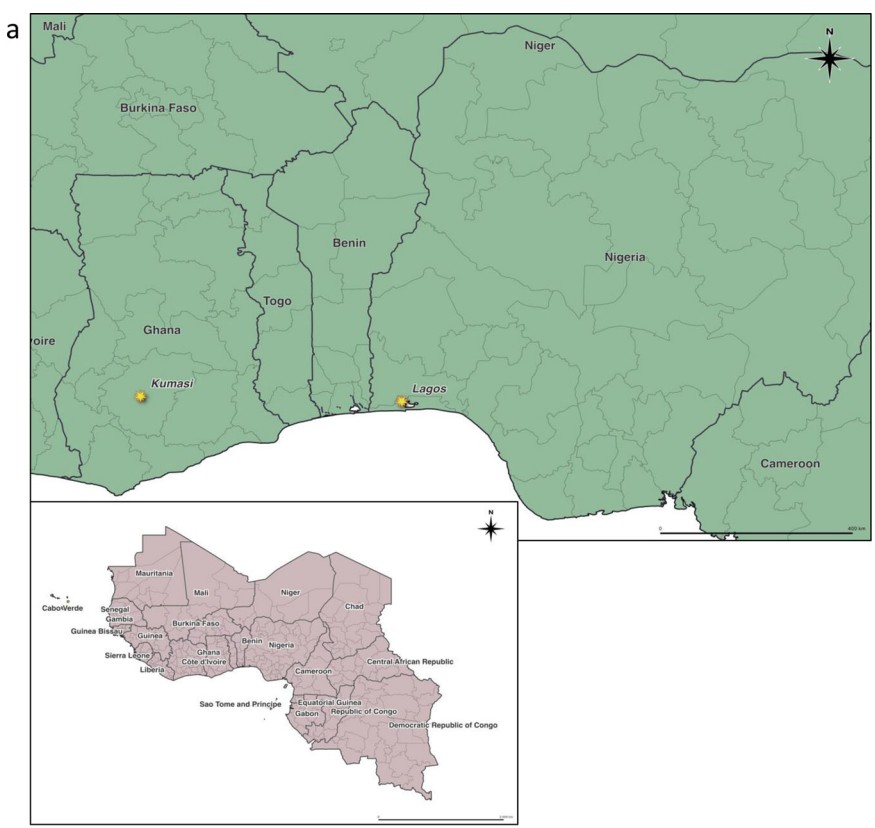

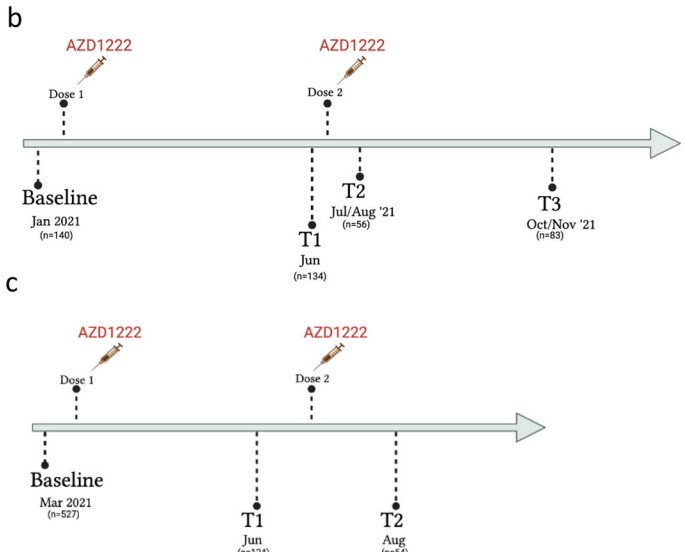

**Fig. 1 | Study sites and design. a** Map of Western Africa showing study sites in Ghana and Nigeria. Study design and flow of patient disposition for recipients of AZD1222 two dose in (**b**) Lagos, Nigeria and (**c**) Kumasi, Ghana cohorts.

**Table 1 | Baseline characteristics of study participants in longitudinal vaccine response study stratified by IgG anti-N status**

| | Follow-up | | | | | |
| --- | --- | --- | --- | --- | --- | --- |
| | Lagos cohort | | | Kumasi cohort | | |
| | All | N-Ab +ve at baseline | N-Ab −ve at baseline | All | N-Ab +ve at baseline | N-Ab −ve at baseline |
| Characteristic | | | | | | |
| Total number, n (%) | 49 (100) | 25 (51) | 24 (49) | 45 (100) | 13 (29) | 32 (71) |
| Age, median years (IQR) | 39 (31, 46) | 37 (31,44) | 40 (30, 49) | 24 (22, 29) | 25 (22, 50) | 24 (22, 27) |
| Male gender, n (%) | 23 (47) | 11 (44) | 12 (50) | 18 (40) | 6 (33) | 12 (67) |
| Serum ID50 (GMT, 95 % CI)[a] | | | | | | |
| Baseline | 145 (95, 224) | 431 (297, 627) | 47 (29, 75) | 57 (40, 79) | 106 (55, 201) | 45 (30, 66) |
| 1 month post 2nd dose | 2579 (1689, 3938) | 4674 (2738, 7981) | 1423 (786, 2577) | 1049 (588, 1871) | 2128 (822, 5513) | 773 (366, 1636) |
| 3 months post 2nd dose | 1695 (1112, 2584) | 2217 (1365, 3600) | 1267 (612, 2620) | N/A | N/A | N/A |

[a]Serum Geometric mean titre (GMT) against Wu-614G wild type virus in (1) Lagos, Nigeria and (2) Kumasi, Ghana. *N/A* not available.

breakthrough rate of 8/49 (16%, Fig. 4 and Supplementary Fig. 5). These individuals also experienced increase in antibodies to S and RBD that mirrored N antibody dynamics (Fig. 4). We were also able to measure binding antibodies to Omicron that were around a log lower in titre as compared to Wu-1 binding antibodies as expected (Fig. 3).

To investigate whether suboptimal immune response was related to subsequent breakthrough, we compared neutralizing antibody titres 1 month post-second dose between those with (n = 8) or without breakthrough infection (n = 15). We found no significant difference in neutralisation between the groups (p = 0.36, Fig. 5a left panel). However, and as expected, neutralizing titres were higher at the last time point in individuals who had experienced vaccine breakthrough infection (with no evidence of infection prior to vaccine), indicating a boosting effect of infection in addition to vaccine with resulting hybrid immunity (Fig. 5a right panel). We noted that the increase in titres against Delta PV observed in breakthrough was significantly greater than the increase for WT and Omicron PVs, coincident with the Delta wave of infection in mid 2021. It is notable that omicron-s1 specific binding antibodies were observed to have increased significantly (p = 0.024) between 1 m and 3 m post second-dose in individuals with breakthrough infection, despite the dominating variant at the time being the delta variant (supplementary Fig. 6); no correlation was found between neutralizing Ab titres to omicron and Omicron s1 binding antibodies at 1 m and 3 m post second-dose (r = 0.18 and r = 0.27 respectively; p = ns).

## Discussion

A pivotal clinical trial for AZD1222 in South Africa (n = 1010 vaccinees) during the Beta wave in mid to late 2020 showed poor protection against infection with Beta, correlating with immune escape in vitro using 19 samples from the vaccine arm[19]. No cases of severe disease or deaths were reported in the placebo or vaccine arms. Since then, AZD1222 has been deployed in more countries than any other vaccine (https://ourworldindata.org/covid-vaccinations), yet there are no real-world data on two dose AZD1222 neutralising antibody responses from Africa. In addition, neither the impact of prior infection nor the impact of infection following vaccination with two dose AZD1222 on neutralising antibody responses have been reported in this setting. One study from Malawi in the pre-Omicron era showed that a single dose of AZD1222 boosted neutralising and binding S antibody responses at 35 days post vaccine in 12 individuals with prior infection. There were no data on waning following vaccination or data on individuals not previously infected[20].

Here, we explored vaccine elicited and infection elicited neutralising antibodies in two west African settings. We first observed high prevalence (44%) of prior SARS-CoV-2 infection in Nigerian HCWs presenting for vaccination in early-2021, as determined by binding anti-N antibodies, with a lower prevalence of 28% in Ghana. There are no population-level SARS-CoV-2 seroprevalence data from west Africa, but a recent modelling derived estimate of 72% in November 2021 was reported in a global analysis[21]. Anti-N Ab titres in some HCWs had declined to below cut-off in our Ab binding assay and detectable neutralisation and anti-S Ab in baseline pre-vaccine samples provided evidence of even higher prevalence of prior infection in this cohort prior to vaccination (>50% in Nigeria and around 40% in Ghana). Our data show that anti-N IgG measurements alone will underestimate past infection by around 50%, requiring our combination approach to accurately measure prevalence. This is an important caveat to the use of assays using N antibody to estimate seroprevalence, as recently suggested in a study comparing commercial assays[22]. Our data suggest that use of Spike derived estimates may be more reliable, though differentiation between infection and vaccination cannot be made. Notably, there was no vaccine available in country during the baseline screening period, and travel abroad was highly restricted. This rules out the possibility that those N negative and S positive had been vaccinated. Although waning of N antibody with rising titre upon re-infection has been reported before[23,24], to our knowledge rigorous assessment of 'occult' past infection revealed by neutralisation and presence of S binding antibodies has not been described in the African setting.

It is important to note that the epidemic control policy in Nigeria and Ghana[25] were relatively homogenous (especially during the early waves of the pandemic when this study was conducted), comprising of a combination of lockdowns, travel restriction and curfews, social distancing, quarantine, robust surveillance mechanisms, contact tracing and public health education[26–28]. Hence the heterogeneity in seroprevalence between the study sites may relate to Nigerian HCWs being at higher risk of exposure than the non HCW population that dominated the Ghanaian study site.

Our second major finding is that two AZD1222 vaccine doses led to a significant increase in neutralisation of WT Wu-1 D614G, Delta and Omicron PV, with GMT in the region of 1000 and higher titres in those who had evidence for prior infection at baseline. Data from high income settings have also shown a similar phenomenon[29,30]. A report from South Africa with single dose Ad26.CoV2.S also demonstrated vaccine boosting of infection-acquired immune responses[31], however, unlike our study, these cited studies were conducted before emergence of the highly immune evasive Omicron variant[2].

Thirdly, 16% of Nigerian participants experienced infection between month 1 and month 3 post vaccination. Neutralising and binding Ab titres 1 month post vaccine, prior to breakthrough, did not appear to be associated with breakthrough infection, although this should be interpreted with caution given the small numbers and the fact that the community/occupational exposure to infection may have been heterogeneous. Those individuals with breakthrough had

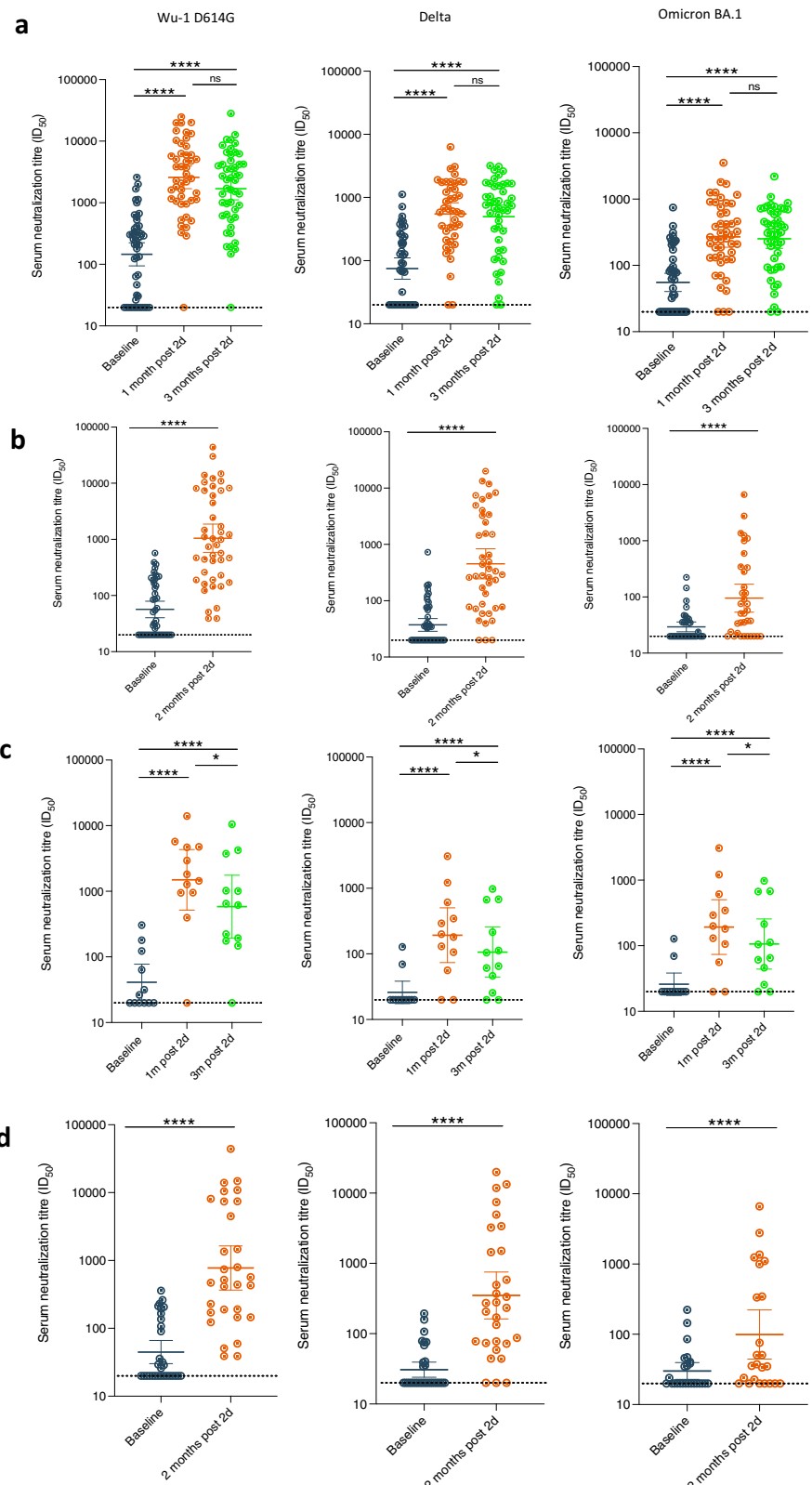

significantly higher plasma neutralisation for Delta PV but the differences for WT and Omicron PVs did not reach statistical significance. This is in keeping with the fact that Delta was circulating in west Africa during summer of 2021[5,23,32], and the breakthrough infections were likely with Delta, explaining the specific increase in neutralisation against this variant. Binding Abs to WT and Omicron S1 protein increased significantly in those with breakthrough infection, though

neutralising GMT for Omicron, whilst higher than in the non-infected, was nonetheless low in this group. One possible explanation for the observation of increased Omicron binding antibodies following breakthrough infection despite the circulating variant of concern during the study period being Delta may relate to the assay targeting the full s1 domain of the spike protein rather than the more specific RBD and thereby allowing non-neutralizing antibodies[33] targeting

**Fig. 2 | Longitudinal SARS-CoV-2 neutralization by sera from AZD1222 vacci-nated individuals in two west African countries. a** Plasma neutralization of pseudovirus after two doses of the AZD1222 from Nigerian participants at three consecutive time points: baseline (prior to first-dose vaccination), 1 mth after 2nd dose vaccination and 3 mth after vaccination (n = 49). **b** Plasma neutralization of pseudovirus after two doses of the AZD1222 from Ghanaian participants at two consecutive time points: baseline (prior to first-dose vaccination) and 1 mth after 2nd dose vaccination (n = 45). Data are representative of two independent experiments comprising of two technical replicates. **c** Plasma neutralization of pseudotyped virus after two doses of the AZD1222 vaccine against VOC from (n = 15) Nigerian participants at baseline (prior to 1st dose vaccination), 1m (1 month) after 2nd dose vaccination and 3m (3 months) after vaccination and were anti-N IgG negative throughout study period. **d** Plasma neutralization of pseudotyped virus after two doses of the AZD1222 vaccine against VOC from (n = 32) in Ghanaian participants at baseline (prior to 1st dose vaccination) and 1m (1 month) after 2nd dose vaccination and were anti-N IgG negative throughout study period. Data points were compared using Wilcoxon test and shown as geometric mean titre (GMT) with 95% CI. Data are representative of two independent experiments comprising of two technical replicates. *P < 0.05; **P < 0.01; ***P < 0.001; ****P < 0.0001; ns = not significant.

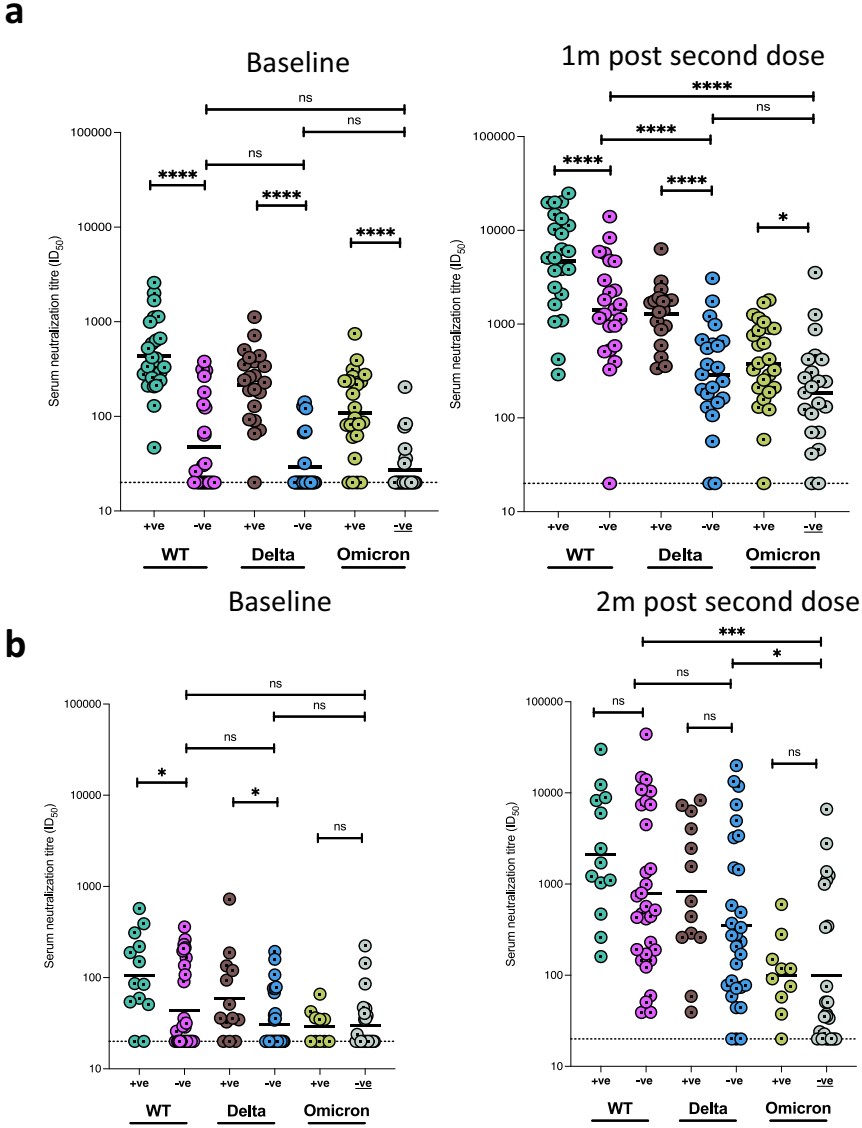

**Fig. 3 | Longitudinal SARS-CoV-2 neutralization by sera from AZD1222 vaccinated individuals in two west African countries stratified by N antibody status at baseline. a** Neutralisation titres before and after vaccine (n = 49) stratified by N antibodies at baseline in Nigerian participants. **b** Neutralisation titres before and after vaccine (n = 45) stratified by N antibodies at baseline in Ghanaian participants. Data points were compared using Wilcoxon test and shown as geometric mean titre (GMT) with 95% CI. Data are representative of two independent experiments comprising of two technical replicates. *P < 0.05; **P < 0.01; ***P < 0.001; ****P < 0.0001; ns = not significant.

epitopes outside of the RBD to dominate the binding antibody assay signal.

The limitations of the study include a modest follow-up period, though we had nearly 100 participants with sequential follow-up data and samples and over 700 baseline samples for binding Ab studies. The underlying community populations of Lagos and Kumasi were not sampled in a systematic way given vaccine delivery was first under-taken in individuals in the health sector; therefore, the findings may not be fully generalisable to the respective countries or region. It is also noteworthy that time point of sampling differed between the two cohorts with the Ghanaian cohort only having a 3 month sampling window in contrast to 6 months for Nigeria.

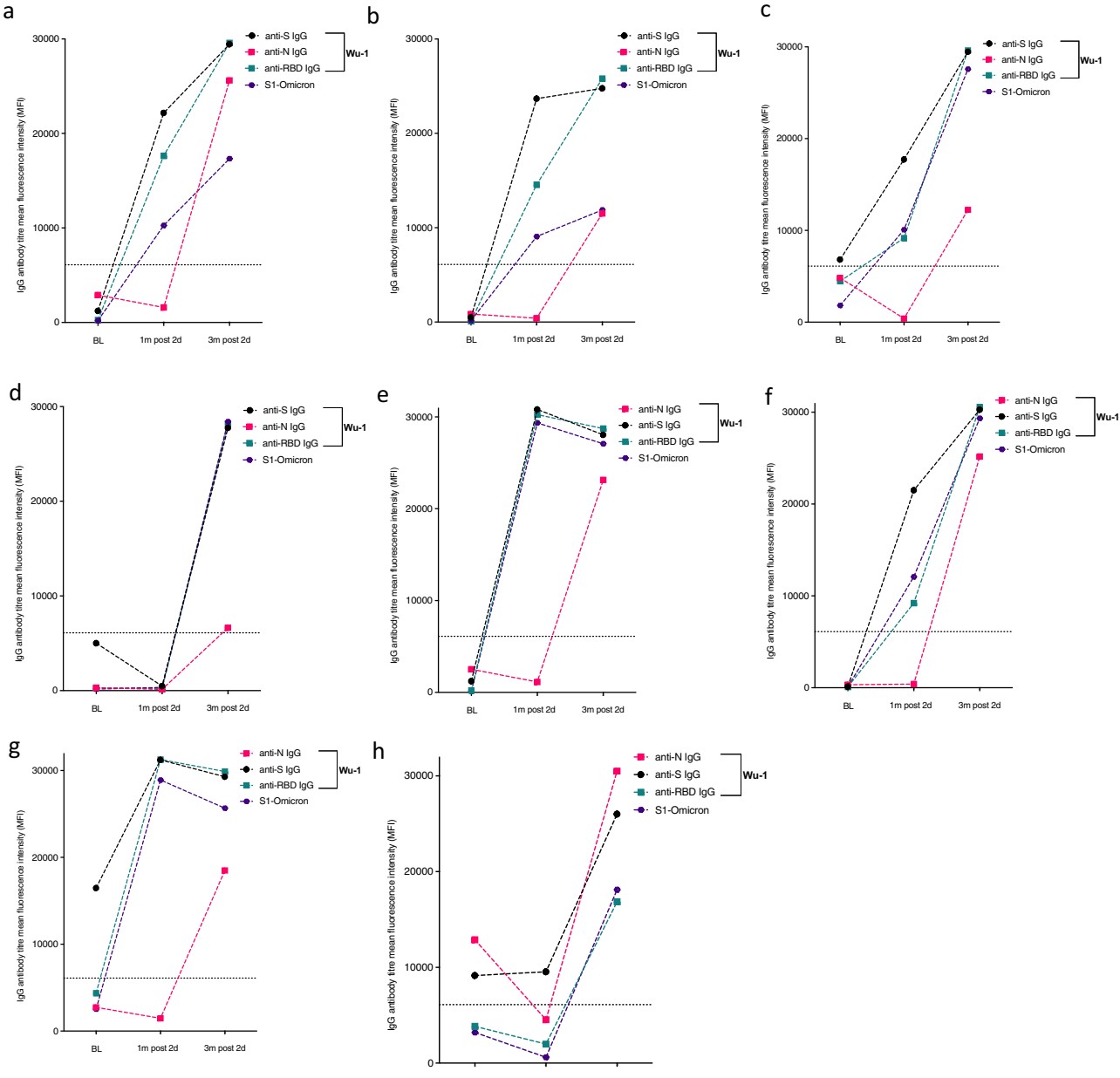

**Fig. 4 | Kinetics of anti-SARS-COV-2-IgG antibodies in eight participants with evidence of breakthrough infection in the Nigerian cohort. a–g** including one (**h**) with evidence of breakthrough infection and re-infection with SARS-COV-2 reinfection following evidence of positive IgG anti-N at baseline; negative IgG anti-N at 1 month after second dose and positive IgG anti-N 3 months post-second dose. Binding antibodies to Wu-1 and Omicron BA.1 are shown.

In addition, we did not measure non-neutralising antibody activities such as ADCC (antibody dependent cellular cytotoxicity) in vaccinees[34], or T cell responses such as secretion of IL-2 or interferon gamma in response to spike protein. T cell responses in particular have been reported to contribute to attenuated disease severity[35] and cellular immunity could explain our lack of association here and in India[3] between neutralising antibody titres and breakthrough infection with Delta. One further consideration relates to cross reactive binding antibody responses with seasonal CoV; this possibility arises due to finding a small fraction of N antibody positive individuals (around 5%) with negative RBD and S antibodies, also observed previously in a UK early pandemic cohort with our assay[36]; however, this observation could also arise due to faster decline of spike specific antibodies compared to N specific antibodies. Due to limited sample volume, we were unable to perform experiments to evaluate cross-reactivity to seasonal coronaviruses.

Finally, we were not able to identify the variants causing breakthrough infections by sequencing of respiratory samples, although neutralisation profiling was consistent with breakthrough infections largely driven by Delta, fitting the epidemiology of variants at the time of sampling.

We conclude that AZD1222 is immunogenic in two independent real world west African cohorts with significantly higher than previously expected background seroprevalence and incidence of breakthrough infection over a short time period. We have shown that N alone is inadequate as a marker for prior infection, and that in unvaccinated populations RBD antibodies increase detection of prior infection, validated by neutralisation activity in sera that are N antibody negative. Prior infection and breakthrough infection induced higher anti-SARS-CoV-2 Ab responses at 3 months post-vaccine against all widely circulating VOCs, with reduced waning. However, plasma neutralisation against Omicron BA.1 was low regardless of prior exposure.

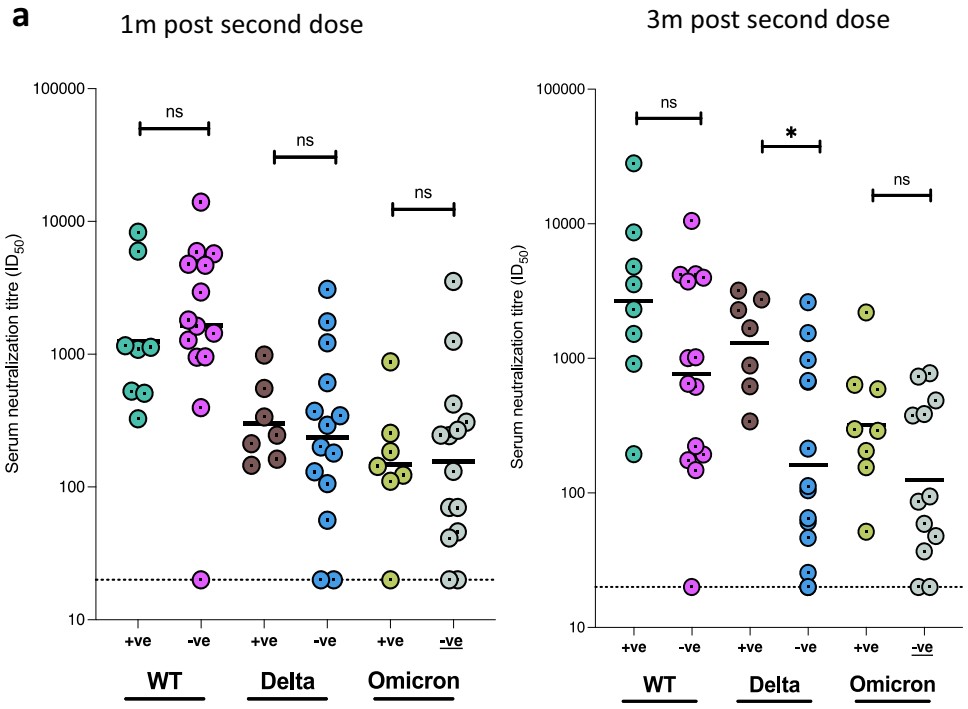

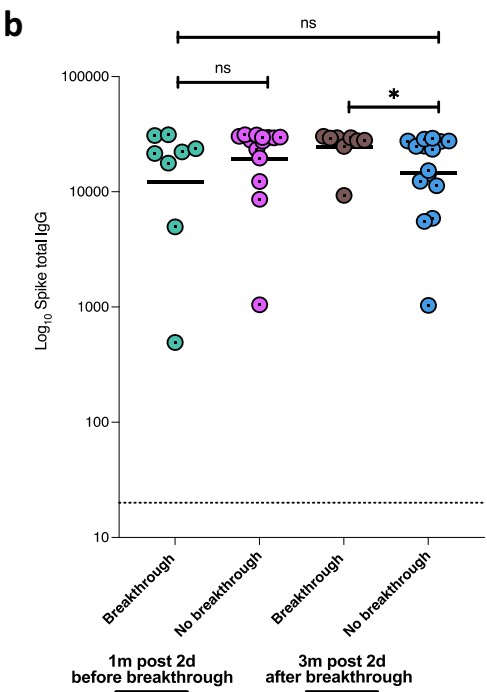

**Fig. 5 | Neutralising and binding SARS-CoV-2 antibody responses one and 3 months after vaccination in context of breakthrough in the Nigerian cohort.** **a** Serum neutralization against pseudotyped virus from individuals with vaccine breakthrough occurring between months 1 and 3 post second dose and with no evidence of previous infection prior to vaccine ($n = 8$, N antibody +ve), and those without breakthrough or past infection prior to vaccine ($n = 15$, N antibody −ve).

**b** Total anti-spike binding IgG levels in individuals with breakthrough infection between 1 and 3 months post vaccination ($n = 8$) and with no evidence of 'natural' infection ($n = 15$). Data points were compared using Wilcoxon test and shown as geometric mean titre (GMT) with 95% CI. Data are representative of two independent experiments comprising of two technical replicates. *$P < 0.05$.

Two-dose AZD1222 has been shown to generate lower neutralisation titres and efficacy across different settings, in comparison to mRNA vaccines[2,3,37,38]. mRNA vaccine 'booster' third doses induce broader, potent responses against Omicron BA.1[2,39] and reduce mortality in the elderly[40]. Therefore, booster dosing after AZD1222 with mRNA vaccine should be considered in the African setting, even after natural infection and hybrid immunity, as future variants may be more pathogenic compared to BA.1[2,41,42]

whilst maintaining immune evasion on a background of waning immunity.

## Methods

### Study population and sampling

**Nigeria.** Health care workers (HCWs) and Health workers (HWs) at the Nigerian Institute of Medical Research (NIMR) and Federal Medical Centre, Ebute Metta, volunteering to be vaccinated with two doses of AZD1222 12 weeks apart were recruited to the study following signed informed consent. HCWs were defined as patient-facing staff such as nursing and midwifery professionals, pharmacists, social workers, and laboratory scientists involved in nasopharyngeal sample collection. HWs were defined as non-patient facing staff such as computing professionals, administrative associate professionals, secretaries, clerks, drivers, and laboratory scientist involved in sample processing. The study design comprised of a prospective longitudinal cohort study of adult patients who were eligible to receive their first-dose vaccination between 13 March 2021 and 31 March 2021 and recruited into the NIMR vaccine effectiveness study. Map of study location is shown in Fig. 1a: Participants provided plasma sample at baseline (prior to first-dose, T0), before second dose (T1), 1 month after second dose (T2) and 3 months after second dose (T3) [Fig. 1b]. Testing was performed on T0, T2 and T3 samples.

**Ghana.** Healthcare workers, university staff, and university students were randomly selected from peripheral vaccination sites in Kumasi, Ghana namely, (i) Kumasi Centre for Collaborative Research in Tropical Medicine (KCCR) (ii) Clinical hostel of the School of Medicine and Dentistry, Kwame Nkrumah University of Science and Technology (KNUST) (iii) Kwadaso Seventh-day Adventist Hospital, Kumasi and (iv) Kumasi South Hospital, Kumasi. The study design also comprised of a prospective longitudinal cohort study of adult patients who were eligible to receive their first-dose vaccination between 3 March 2021 and 11 March 2021 and prospectively recruited into the study. Participants were enroled based on their willingness to be vaccinated with two doses of AZD1222 8–12 weeks apart following signed informed consent. Participants provided plasma prior to vaccination (T0), 8–12 weeks after the first dose (T1) and >8 weeks after second dose (T2). [Fig. 1c]. Testing was performed on T0 and T2 samples.

**Laboratory methods and sample testing.** Binding IgG antibodies (Abs) against SARS-COV-2 receptor-binding domain (RBD), trimeric spike protein (S) and nucleocapsid protein (N) were measured using the Luminex-based SARS-CoV-2-IgG assay by flow cytometry as previously detailed[1,43]. We defined positive total anti-S antibody (anti-S) as anti-S IgG above cut-off of 1896 mean fluorescence intensity (MFI) and positive RBD as anti-RBD above cut-off of 456 mean fluorescence intensity (MFI). Cut-offs were defined based on analysis of 'true' positive (convalescent) and negative pre-pandemic samples. We defined previous SARS-CoV-2 infection as positive anti-N IgG above a cut-off of 6104 mean fluorescence intensity (MFI) or negative anti-N IgG with positive anti-RBD. Vaccine breakthrough infection was defined as the absence of IgG anti-N at 1 month post-second dose and its presence at 3 months post second-dose.

For plasma neutralising antibody measurement, SARS-CoV-2 lentiviral pseudovirus (PV) were prepared by transfecting HEK293T cells with Wu-1-614G wild type (WT), B.1.617.2 (Delta) and BA.1 (Omicron) plasmids in conjunction with p8.91 HIV-1 gag-pol expression vector[9,18]. We and others previously showed high correlation between PV and live virus neutralisation[17,44]. Sample testing was performed at baseline, T2 and T3 time points. Pseudovirus neutralisation was performed on Hela-ACE2 cells using SARS-CoV-2 spike PV expressing luciferase. Briefly, plasma samples were heat inactivated at 54 °C for 1 h, serially diluted in duplicates and incubated with PVs at 37 °C for 1 h prior to addition of Hela-ACE2 cells[4]. Plasma dilution/virus mix were incubated for 48 h in a 5% $CO_2$ environment at 37 °C, and luminescence was measured using Bright-Glo Luciferase assay system (Promega). All neutralisation assays were repeated in two independent experiments containing two technical replicates for each condition. Neutralisation was calculated relative to virus-only controls as a mean neutralisation with s.e.m. Half maximum inhibitory dose ($ID_{50}$) was calculated in GraphPad Prism version 9.3.1 and $ID_{50} > 20$ was considered positive. 293 T cells ATCC: CRL-3216 and HELA-ACE2 cells were a kind gift from Dr. James Voss, SCRIPPS.

**Statistical analysis.** Geometric Mean Titre (GMT) with 95% confidence interval (CI) of neutralisation antibody was calculated across time points. Characteristics of participants were expressed as proportions and percentages for categorical variables and median inter-quartile range (IQR) for continuous variables. Mann–Whitney or Wilcoxon test was used to compare neutralisation antibody titres across time points and compare participants based on IgG anti-N strata. Differences between neutralisation antibody titres in IgG anti-N participants with $ID_{50} > 20$ were compared by Kruskal Wallis non-parametric test. Statistical analysis was performed using GraphPad Prism version 9.3.1.

### Reporting summary

Further information on research design is available in the Nature Research Reporting Summary linked to this article.

## Data availability

All data generated or analysed in this study are included in this published article and its Supplementary Information file. Source Data are provided with this article.

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

## Acknowledgements

We are thankful to the volunteers who participated in the study. A.A. is supported by Africa Research Excellence Fund Research Development Fellowship (AREF-318-ABDUL-F-C0882) and Cambridge-Africa award. A.A. and S.A.K. are supported by Bill and Melinda Gates Foundation via the Phylogenetics and Networks for Generalised Epidemics in Africa (PANGEA) (grant number OPP1175094). M.O., A.S. and R.O.P. were supported by grant from the Government of Ghana through the World Bank. R.K.G. is supported by a Wellcome Trust Senior Fellowship in Clinical Science (WT108082AIA).

## Author contributions

Study conception and design: A.A., D.O., M.O., G.O., O.E., Ri.A., Ro.A., D.F., R.O.P., P.M., B.L.S. and R.K.G.; data collection: A.A., D.O., A.S., A.A.S., Ru.A., A.O., H.W.A., J.S.K., A.D., S.M., O.U., G.A., M.O., K.O., R.A., G.L., O.O., K.O.B., J.O.F., L.D.A., N.A.K.-B., J.S.K., E.A., J.A., D.B.K., D.O.O.; performed experiments: A.A., S.A.K., S.E., L.C.G., I.A.T.M., B.M., R.D.; data analysis: A.A., S.A.K., D.F., J.A.,C.O., F.I., S.E., L.C.G., R.D., I.A.T.M., B.M., S.H.A. and R.K.G.; data interpretation: A.A., S.A.K., M.O., D.O., Ri.A., Ro.A., J.A., A.S., F.I., R.D., P.M., R.O.P. and R.K.G.; manuscript preparation; A.A. and R.K.G. wrote the first draft of the manuscript with the critical input of all co-authors. All authors reviewed the results and approved the final version of the manuscript.

## Competing interests

R.K.G has received honoraria for educational activities from Janssen, Moderna, and GSK.

## Ethics

This study was approved by the Institutional Review Board of NIMR (IRB-21-040) and the Committee of Human Research, Publication and Ethics of KNUST (CHRPE/AP/091/21). All participants provided written informed consent.

## Additional information

Adam Abdullahi[1,2,3,15], David Oladele[4,15], Michael Owusu[5,6,15], Steven A. Kemp[1,2,15], James Ayorinde[4], Abideen Salako[4], Douglas Fink[7,8], Fehintola Ige[4], Isabella A. T. M. Ferreira[1,2], Bo Meng[1,2], Augustina Angelina Sylverken[5,6], Chika Onwuamah[4], Kwame Ofori Boadu[9], Kazeem Osuolale[4], James Opoku Frimpong[5], Rufai Abubakar[4], Azuka Okuruawe[4], Haruna Wisso Abdullahi[3], Gideon Liboro[4], Lawrence Duah Agyemang[10], Nana Kwame Ayisi-Boateng[5], Oluwatosin Odubela[4], Gregory Ohihoin[4], Oliver Ezechi[4], Japhet Senyo Kamasah[5], Emmanuel Ameyaw[5], Joshua Arthur[10], Derrick Boakye Kyei[6], Dorcas Ohui Owusu[5], Olagoke Usman[11], Sunday Mogaji[11], Adedamola Dada[11], George Agyei[12], Soraya Ebrahimi[13], Lourdes Ceron Gutierrez[13], Sani H. Aliyu[13], Rainer Doffinger[13], Rosemary Audu[4], Richard Adegbola[4], Petra Mlcochova[1,2]✉, Richard Odame Phillips[5,6]✉, Babatunde Lawal Solako[4]✉ & Ravindra K. Gupta[1,2,14]✉

[1]Cambridge Institute of Therapeutic Immunology & Infectious Disease (CITIID), Cambridge, UK. [2]Department of Medicine, University of Cambridge, Cambridge, UK. [3]Institute of Human Virology, Abuja, Nigeria. [4]Nigeria Institute of Medical Research (NIMR), Yaba, Lagos, Nigeria. [5]Kwame Nkrumah University of Science and Technology, Kumasi, Ghana. [6]Kumasi Centre for Collaborative Research in Tropical Medicine, Kumasi, Ghana. [7]Faculty of Infection and Tropical Diseases, London School of Hygiene and Tropical Medicine, London, UK. [8]Department of Infection and Immunity, University College London, London, UK. [9]Kumasi South Hospital, Kumasi, Ghana. [10]Komfo Anokye Teaching Hospital, Kumasi, Ghana. [11]Federal Medical Centre, Ebutte Metta, Lagos, Nigeria. [12]Kwadaso Seventh Day Adventist Hospital, Kumasi, Ghana. [13]Addenbrooke's Hospital, Cambridge University Hospitals NHS Foundation Trust, Cambridge Biomedical Campus, Cambridge, UK. [14]Africa Health Research Institute, Durban, South Africa. [15]These authors contributed equally: Adam Abdullahi, David Oladele, Michael Owusu, Steven A. Kemp. ✉e-mail: pm685@cam.ac.uk; phillips@kccr.de; tundesalako@hotmail.com; rkg20@cam.ac.uk

