## [Peer Review File · Nature Communications]

REVIEWER COMMENTS

Reviewer #1 (Remarks to the Author):

In this manuscript, Adam Abdullahi and colleagues measured baseline SARS-CoV-2 seroprevalence and levels of protective antibodies prior to and post AZD2222 vaccination, as well as neutralizing antibody responses to VOCs in two West African cohorts. They found that anti-N but not anti-S IgG measurement alone missed about 50% of prior infection. Two doses of AZD1222 vaccination led to a significant increase in neutralizing antibody titers. Overall, this study provided limit novel information and some critical concerns were also raised.

1. The immunogenicity and efficacy of AZD1222 vaccine have been widely reported. If the authors would like to focus on the breakthrough infection following AZD1222 vaccination, the manuscript should be condensed and revised substantially.
2. Cohorts from two countries at different geographical location in West Africa were enrolled in this study. Did the different epidemic prevention and control policies influence the baseline seroprevalence?
3. As the authors concluded SARS-CoV-2 N specific antibody detection was inaccurate and improper to distinguish prior infection. However, it was still broadly used to evaluate prior infection in this study.
4. N protein is more abundantly expressed in CoV infections. Did the authors find any individuals that were positive for N-specific antibodies, but negative for RBD-specific antibodies? Any cross reactivity was observed with seasonal CoVs?
5. To confirm breakthrough infection, SARS-CoV-2 nucleic acid test must be done, rather than only determined by serological tests.
6. The statistical method used in Figure 2 and Figure 5 should be provided in their legends.
7. Several grammatic errors and typos were noted.

Reviewer #2 (Remarks to the Author):

This is a novel, significant and important body of work. As the authors say, the AZ vaccine has done quite a large amount of the heavy-lifting in a global health context, yet there is a paucity of real-life immunology data. This study includes rather comprehensive analysis of 140 Nigerian HCW and a cohort of 527 in Ghana. By looking at S RBD and N serology as well as pseudotype virus neutralisation against WT, Delta and Omicron, the authors are able to describe responses at baseline, then at 1 and 3 months after 2nd dose. This enables them to look both at waning and at possible breakthrough infections.

The paper is clearly written and a pleasure to read.

Some might regard the title and abstract a little bland and non-committal considering the various messages from the data.

While national surveillance data from Nigeria and Ghana across these periods may be limited, it would be important to offer the reader a little more context for the backdrop/context for this study locally at that time.

Does the N serology allow the authors to conduct any comparison of immunity through hybrid priming?

Reviewer #3 (Remarks to the Author):

In their study, "SARS-CoV-2 Antibody Responses to AZD1222 Vaccination in West Africa,"

Abdullahi, Oladele, Owusu, Kemp, and colleagues report the pre-vaccine sero-status and post-vaccine neutralizing antibody levels against SARS-CoV-2 in two West African cohorts: health care workers in Lagos, Nigeria, and healthcare and university-affiliated individuals from Kumasi, Ghana. The authors successfully recruited an impressive number of individuals for baseline sero-status testing pre-vaccine, and obtained a follow-up blood draw from a smaller but still substantial number of individuals in each cohort. The manuscript is well-written and the methodology is sound. The data visualization is strong and the data is properly analyzed and displayed for ease of interpretation, with appropriate use of statistics throughout. The authors are measured in their interpretation of the data, come to reasonable conclusions from the data presented, and acknowledge the limitations of the study, namely that the study covers only a modest follow-up period post-vaccination on a cohort of limited demographic diversity, does not include analysis of booster vaccination, and that only neutralizing antibodies and not cellular immune responses or non-neutralizing antibody activities were measured. These limitations are offset by the strengths of the manuscript, which performs a high-quality analysis on a severely understudied population. Although all of the main findings reported here reflect observations made following AZD1222 vaccination in other settings, the authors rightfully contend that the novelty of the work lies in such observations never being made in these populations in West Africa. It is of great importance for the fields of medicine and biomedicine that observations made in European and American populations living near major academic centers be repeated in other settings rather than assuming all findings will apply to all human populations. The authors have made a commendable effort in performing such a study to a high standard of quality and rigor.

The study presents several interesting results, confirming observations in other settings. Participants with evidence of prior SARS-CoV-2 infection had higher baseline neutralizing activity and achieved higher levels of neutralization after the vaccine than individuals with no evidence of prior infection. Antibody levels also waned more gradually, if at all, in those who were previously infected. The data in the study support the well-founded observation that the omicron variant, and to a lesser extent the delta variant, is neutralized less efficiently following vaccination. Data on omicron neutralization, while not novel, are timely given the prevalence of omicron-derived variants in circulation at the time of this review. The study also highlights a number of breakthrough infections, showing clear boosting of pre-existing antibody levels following infection. The small number of infections observed suggest that the neutralizing antibody levels achieved by two-dose vaccination with AZD1222 in this cohort were insufficient to protect against Delta infection, but the authors rightfully do not overinterpret the data from the small number of cases. The authors also performed a general assessment of the frequency of previous SARS-CoV-2 infection in these populations prior to vaccination in early 2021, providing valuable epidemiological information and arguing that measures beyond anti-N antibodies are required for properly assessing prior infection history. Their observations of waning anti-N levels during the study period strongly support this claim.

The novelty is largely restricted to the setting in which the study was conducted and the study is of limited scope, providing data on only neutralizing antibodies, when other aspects of adaptive immunity may also be important in these populations. Nonetheless, the authors have conducted a study that, overall, is deserving and of suitable quality for publication.

Primary concerns

- It may be of interest for the authors to comment on the individuals in Figure 4 for whom omicron-binding antibodies appeared to be boosted to the same degree as Wuhan-binding antibodies following what is presumed to be a Delta virus infection. Did Omicron neutralization following breakthrough infection (Figure 5) correlate with these Omicron-binding antibodies, or to the change in antibody titre? If not, this may be due to the use of the full S1 domain in Figure 4 rather than only the Omicron RBD, which could allow non-neutralizing antibodies targeted unmutated epitopes outside the RBD to dominate the signal in the assay.

Minor points

- Though it is not reasonable to ask the authors to measure these responses in this cohort, the authors should comment in the Discussion on the importance of cellular immune responses in addition to antibodies. The observation that breakthrough infections were not correlated with neutralizing antibody levels suggests that the levels of antibody achieved in these cohorts were

insufficient to prevent infection against Delta, and presumably Omicron. Thus, vaccine-induced memory B cells and T cells may play an outsized role in mediating protection in this setting.

Additional comments to benefit the authors

1. I think the legend for Figure 2b should read "from Ghanaian participants at two consecutive time points" rather than "at three consecutive time points."
2. The title for "Baseline" in Figure 3b is off-center.
3. In the 12th line of the section titled Longitudinal neutralizing and binding antibody responses following vaccination, I think the reference to Figure 2a should instead be to Figure 2b.

RESPONSE TO REVIEWER'S COMMENTS

Reviewer 1:

1. The immunogenicity and efficacy of AZD1222 vaccine have been widely reported. If the authors would like to focus on the breakthrough infection following AZD1222 vaccination, the manuscript should be condensed and revised substantially.

Response: It is agreed that the immunogenicity and efficacy of AZD1222 vaccine has been widely reported in high income non-African settings. There is limited evidence on immunogenicity of two dose AZD1222 vaccination in the African setting, and no published data outside clinical trials as far as we know. Similarly, there are little or no high quality data on pre vaccine seropositivity and prior infection for west Africa. The novelty of this research work is firstly a thorough assessment of prior exposure and use of N as a sole marker, secondly real world neutralising antibody response data from populations in West African using multi pronged assays in the context of prior infection and hybrid immunity, and finally uniquely detailed data on breakthrough infections post vaccination, showing v high burden of infection during the Delta wave.

2. Cohorts from two countries at different geographical location in West Africa were enrolled in this study. Did the different epidemic prevention and control policies influence the baseline seroprevalence?

Response: We thank the reviewer for highlighting this important point. We have now indicated in the discussion that the epidemic control policy in Nigeria and Ghana, especially during the early waves of the pandemic were relatively homogenous, comprising of a combination of lockdowns, travel restriction and curfews, social distancing, quarantine, robust surveillance mechanism, contact tracing and public health education. Hence, the observation of heterogeneity in seroprevalence is more likely due to Nigerian HCWs been at higher risk of exposure than participants in Ghana. We have provided additional information on pandemic conditions in west Africa as supplementary text.

3. As the authors concluded SARS-CoV-2 N specific antibody detection was inaccurate and improper to distinguish prior infection. However, it was still broadly used to evaluate prior infection in this study.

Response: The standard marker for serologically evaluating previous infection is the IgG anti-N, using ELISA. We employed the use of flow-cytometric based analysis of binding antibodies to N as a start, based on prior use of this marker in the field. We added other markers, including IgG anti-RBD in order to evaluate accuracy of N as a marker of prior infection. Importantly we use neutralisation assays to corroborate findings from serology. The combination of both anti-N and anti-RBD to evaluate previous infection in real world cohorts is novel for the African setting, and bolstered in this report by in vitro neutralisation experiments using a range of VOC.

4. N protein is more abundantly expressed in CoV infections. Did the authors find any individuals that were positive for N-specific antibodies, but negative for RBD-specific antibodies? Any cross reactivity was observed with seasonal CoVs?

Response: We thank the reviewer for making this important point. Indeed N is well conserved across CoV and highly expressed during infection. Due to limited sample volume, we were unable to perform experiments to evaluate cross-reactivity to seasonal coronaviruses. Discussion on cross reactivity with CoV has now been included in the limitations section, although waning kinetics are also a plausible explanation for discrepancies between N and spike antibodies in unvaccinated individuals:

‘One further consideration relates to cross reactive binding antibody responses with seasonal CoV; this possibility arises due to finding a small fraction of N antibody positive individuals (around 5%) with negative RBD and S antibodies, also observed previously in a UK early pandemic cohort with our assay³⁷; however, this observation could also arise due to faster decline of spike specific antibodies compared to N specific antibodies. Due to limited sample volume, we were unable to perform experiments to evaluate cross-reactivity to seasonal coronaviruses. ‘

5. To confirm breakthrough infection, SARS-CoV-2 nucleic acid test must be done, rather than only determined by serological tests.

Response: Although the study was longitudinal and prospective, sampling was performed within the context of what was technically feasible (collecting only plasma samples), especially as local policy focus was on vaccine rollout. Nucleic acid testing would have

required routine collection and testing of pharyngeal swabs from a large number of participants, and resources/infrastructure were not in place to do this. The Figure below, which combines anti-S, anti-RBD and anti-N provides good evidence nonetheless, that breakthrough infections had occurred following two-doses of vaccination. We can happily provide similar figures to those below for individuals without breakthrough to emphasise the specificity of use of binding antibodies to detect breakthrough.

6. The statistical method used in Figure 2 and Figure 5 should be provided in their legends.

Response: This has now been corrected, thank you for pointing this out.

7. Several grammatical errors and typos were noted.

Response: The paper has now been extensively reviewed and we apologise for any errors.

Reviewer 2:

This is a novel, significant and important body of work. As the authors say, the AZ vaccine has done quite a large amount of the heavy lifting in a global health context, yet there is a paucity of real-life immunology data. This study includes rather comprehensive analysis of 140 Nigerian HCW and a cohort of 527 in Ghana. By looking at S RBD and N serology as well as pseudotype virus neutralisation against WT, Delta and Omicron, the authors are able to describe responses at baseline, then at 1 and 3 months after 2nd dose. This enables them to look both at waning and at possible breakthrough infections.

The paper is clearly written and a pleasure to read.

Some might regard the title and abstract a little bland and non-committal considering the various messages from the data.

While national surveillance data from Nigeria and Ghana across these periods may be limited, it would be important to offer the reader a little more context for the backdrop/context for this study locally at that time.

Does the N serology allow the authors to conduct any comparison of immunity through hybrid priming?

Response: We thank the reviewer for the positive comments. The abstract has been modified and we kept the title short and descriptive, though we would welcome suggestions from the reviewer or editor.

Response: We have included a backdrop to the study as follows in the discussion: ‘...epidemic control policy in Nigeria and Ghana were relatively homogenous (especially during the early waves of the pandemic when this study was conducted), comprising of a combination of lockdowns, travel restriction and curfews, social distancing, quarantine, robust surveillance mechanisms, contact

tracing and public health education. Hence the heterogeneity in seroprevalence between the study sites may relate to Nigerian HCWs being at higher risk of exposure than the non HCW population that dominated the Ghanaian study site.'

In addition we have provided supplemental text with information on COVID-19 in Nigeria and Ghana.

Does the N serology allow the authors to conduct any comparison of immunity through hybrid priming?

Response: Yes, we were able to compare immune responses in those with or without prior infection. Anti-N antibody analysis allowed us to perform comparison of immunity through hybrid priming which showed that participants with previous infection had higher levels of neutralizing antibodies after two doses as shown in Figure 3 below.

These responses did not wane at 3 months in those with prior infection (Supp Figure 2a), in contrast to those without prior infection (figure 2C). Furthermore those who had a breakthrough infection also had higher antibody responses at the end of the follow up at 3 months post vaccine (Figure 5b).

Reviewer #3 (Remarks to the Author):

In their study, "SARS-CoV-2 Antibody Responses to AZD1222 Vaccination in West Africa," Abdullahi, Oladele, Owusu, Kemp, and colleagues report the pre-vaccine sero-status and post-vaccine neutralizing antibody levels against SARS-CoV-2 in two West African cohorts: health care workers in Lagos, Nigeria, and healthcare and university-affiliated individuals from Kumasi, Ghana. The authors successfully recruited an impressive number of individuals for baseline sero-status testing pre-vaccine and obtained a follow-up blood draw from a smaller but still substantial number of individuals in each cohort. The manuscript is well-written, and the methodology is sound. The data visualization is strong, and the data is properly analysed and displayed for ease of interpretation, with appropriate use of statistics throughout. The authors are measured in their interpretation of the data, come to reasonable conclusions from the data presented, and acknowledge the limitations of the study, namely

that the study covers only a modest follow-up period post-vaccination on a cohort of limited demographic diversity, does not include analysis of booster vaccination, and that only neutralizing antibodies and not cellular immune responses or non-neutralizing antibody activities were measured. These limitations are offset by the strengths of the manuscript, which performs a high-quality analysis on a severely understudied population. Although all of the main findings reported here reflect observations made following AZD1222 vaccination in other settings, the authors rightfully contend that the novelty of the work lies in such observations never being made in these populations in West Africa. It is of great importance for the fields of medicine and biomedicine that observations made in European and American populations living near major academic centres be repeated in other settings

rather than assuming all findings will apply to all human populations. The authors have made a commendable effort in performing such a study to a high standard of quality and rigor.

The study presents several interesting results, confirming observations in other settings. Participants with evidence of prior SARS-CoV-2 infection had higher baseline neutralizing activity and achieved higher levels of neutralization after the vaccine than individuals with no evidence of prior infection. Antibody levels also waned more gradually, if at all, in those who were previously infected. The data in the study support the well-founded observation that the omicron variant, and to a lesser extent the delta variant, is neutralized less efficiently following vaccination. Data on omicron neutralization, while not novel, are timely given the prevalence of omicron-derived variants in circulation at the time of this review. The study also highlights a number of breakthrough infections, showing clear boosting of pre-existing antibody levels following infection. The small number of infections observed suggest that the neutralizing antibody levels achieved by two-dose vaccination with AZD1222 in this cohort were insufficient to protect against Delta infection, but the authors rightfully do not overinterpret the data from the small number of cases. The authors also performed a general assessment of the frequency of previous SARS-CoV-2 infection in these populations prior to vaccination in early 2021, providing valuable epidemiological information and arguing that measures beyond anti-N antibodies are required for properly assessing prior infection history. Their observations of waning anti-N levels during the study period strongly support this claim.

The novelty is largely restricted to the setting in which the study was conducted, and the study is of limited scope, providing data on only neutralizing antibodies, when other aspects of adaptive immunity may also be important in these populations. Nonetheless, the authors have conducted a study that, overall, is deserving and of suitable quality for publication.

Primary concerns

- It may be of interest for the authors to comment on the individuals in Figure 4 for whom omicron-binding antibodies appeared to be boosted to the same degree as Wuhan-binding antibodies following what is presumed to be a Delta virus infection. Did Omicron

neutralization following breakthrough infection (Figure 5) correlate with these Omicron-binding antibodies, or to the change in antibody titre? If not, this may be due to the use of the full S1 domain in Figure 4 rather than only the Omicron RBD, which could allow non-neutralizing antibodies targeted unmutated epitopes outside the RBD to dominate the signal in the assay.

Response: We have performed additional analyses now visualised in supplementary figure 6 where we compare s1-specific omicron antibodies in individuals with breakthrough infection and perform correlation analysis between s1-specific omicron antibodies and neutralizing antibodies to Omicron. Appropriate discussion has also been added to this and we agree with the reviewer that there is the possibility that non- neutralizing antibodies targeting unmutated epitopes outside the RBD may be dominating the signal in the assay .

Minor points

- Though it is not reasonable to ask the authors to measure these responses in this cohort, the authors should comment in the Discussion on the importance of cellular immune responses in addition to antibodies. The observation that breakthrough infections were not correlated with neutralizing antibody levels suggests that the levels of antibody achieved in these cohorts were insufficient to prevent infection against Delta, and presumably Omicron.

Thus, vaccine-induced memory B cells and T cells may play an outsized role in mediating protection in this setting.

Response: we thank the reviewer for this important point. We have included a discussion point on the possible role of vaccine-induced memory B cells and T cells may play a significant role in mediating protection in this setting.

'In addition, we did not measure non-neutralising antibody activities such as ADCC (antibody dependent cellular cytotoxicity) in vaccinees³⁵, or T cell responses such as secretion of IL-2 or interferon gamma in response to spike protein. T cell responses in particular have been reported to contribute to attenuated disease severity³⁶ and cellular immunity could explain our lack of association here and in India³ between neutralising antibody titres and breakthrough infection with Delta.'

Additional comments to benefit the authors

1. I think the legend for Figure 2b should read "from Ghanian participants at two consecutive time points" rather than "at three consecutive time points."

Response: We thank the reviewer for pointing this out

2. The title for "Baseline" in Figure 3b is off-centre.

Response: We thank the reviewer for pointing this out

3. In the 12th line of the section titled Longitudinal neutralizing and binding antibody responses following vaccination, I think the reference to Figure 2a should instead be to Figure 2b.

Response: We thank the reviewer for pointing this out